# Fixed-Time Formation Control for Unmanned Surface Vehicles with Parametric Uncertainties and Complex Disturbance

Helong Shen [1]  , Yong Yin [1],* and Xiaobin Qian [2]

1   Key Laboratory of Marine Dynamic Simulation and Control, Department of Navigation, Dalian Maritime University, Dalian 116026, China
2   Zhilong (Dalian) Marine Technology Co., Ltd., Dalian 116020, China
*   Correspondence: bushyin@dlmu.edu.cn

**Abstract:** In this paper, under parametric uncertainties and complex disturbances, a leader–follower formation control strategy based on accurate disturbance observer (ADO) and a novel fixed-time fast terminal sliding mode (FTFTSM) control for unmanned surface vehicles (USVs) is proposed. The main contributions of this paper are: (1) A novel fixed-time fast terminal sliding mode tracking control (FTFTSM-TC) strategy is designed for the tracking control subsystem, which greatly improves the convergence rate of the leader USV in trajectory tracking. (2) An ADO is designed to observe lumped disturbances with the smallest approximation error. The ADO greatly reduces the interference of disturbances and improves the performance of the formation system. (3) An ADO-based fixed-time formation control (ADO-FTFC) strategy is developed for the formation control subsystem to maintain the desired formation. Stability of the formation control system is established by the Lyapunov theory. Simulation results show that the proposed control strategy is superior for the USVs formation control.

**Keywords:** unmanned surface vehicles; leader–follower formation; fixed-time theory; sliding mode control; accurate disturbance observer

## 1. Introduction

Recently, USVs have been extensively deployed in both civilian and military sectors due to their effectiveness and efficiency [1,2]. USVs have the ability of performing tasks around the clock, especially in harsh marine environments replacing humans in performing dangerous, time-consuming and laborious tasks. In the recent years, intelligent control of USVs has become a hot topic. Maritime authorities have vigorously promoted the development of USVs, and fruitful research results in the development of a single USV have been reported [3]. However, a single USV cannot meet the demand of complex tasks and changing landscape in real-world marine environments. Therefore, USV formation control has become a hot issue in the field of cooperative control. The USV formation control method designed in this paper can achieve the precise formation control effect and has far-reaching practical significance, for example, USV formation cruises in the military field and personnel search and rescue in the civil field.

As an integral part of cooperative control, the formation control of USVs can not only significantly reduce manpower costs but also enhance the fault tolerance of task completion in unknown marine environments, including disturbances of wind, waves and currents. There are many typical formation control methods, such as the virtual structure method [4], behavior-based control [5], the leader–follower formation control method [6], graph theory [7] and the artificial potential field method [8]. Among these methods, the control rule of the leader–follower formation control strategy is the simplest. Moreover, it translates the USV formation control problem into a USV trajectory tracking problem. As such, we only need to know the leader's motion parameters and the follower's tracking strategy to achieve formation control. Inspired by [4], the leader–follower control method is used to simplify the design process of USV formation control strategies in this work.

In USV formation control, the convergence rate is a key parameter used to evaluate the performance of the formation control system. Typical control methods used in formation system are sliding mode control, parameter adaptation, neural networks, fuzzy logic, and backstepping. Compared with other methods, backstepping has a good control effect for systems with uncertain parameters. However, the ability to resist external disturbance is weak. The sliding mode control is the simplest and most effective, and it has strong robustness for external disturbances and unmodeled dynamics [9]. Therefore, this formation control strategy has been widely used for USVs. Common sliding mode technologies mainly include the integral sliding mode (ISM) [10], the traditional asymptotic convergent sliding mode [11], the nonsingular terminal sliding mode [12] and the terminal sliding mode (TSM) [13]. In [14,15], a non-singular integral sliding mode tracking control strategy combine with adaptive control is proposed for nonlinear systems with disturbances, which can ensure the convergence in finite time, and the effectiveness of the proposed algorithm is verified by experiments. In [16], a non-singular terminal sliding mode tracking control strategy combined with adaptive control is proposed for perturbed nonlinear systems, which has a good control effect. In [17], a composite nonlinear feedback controller is designed for robust tracking, which keep precise tracking control with output saturation. The advantage of the integral sliding mode is the existence of singularity, while the disadvantage of the terminal sliding mode is slow convergence rate and depends on the initial system state. At the same time, the methods described above can only guarantee the convergence of the system in a finite time, and the maximum convergence time of the system has a great impact on the initial state of the system and cannot guarantee a fast and stable convergence rate. In order to alleviate the disadvantages of existing works and improve the convergence rate and stability of the formation system, a fixed-time fast terminal sliding mode (FTFTSM) is developed.

In order to ensure fast response of the formation system, the idea of finite-time control is adopted for trajectory tracking and cluster control of USVs [18–20]. However, the finite-time control dependents on the system initial state. As an extension of finite-time control, which is not affected by the initial system state, fixed-time control is introduced into the multi-agent control [21,22]. As proposed firstly in [23], using the fixed-time control algorithm, the upper bound of the convergence time can be calculated without relying on the initial system state. Therefore, in order to improve convergence rate of the formation system, the fixed-time control strategy is adopted in this paper.

In a practical environment, the uncertainties and complexity of the marine environment must be considered in the design of USV formation control algorithms. How to identify internal unmodeled dynamics and external disturbances of the system quickly and accurately has become a hot issue in USV formation control research. Many researchers are frequently confronted with the challenge of dealing with complex unknown disturbances [24]. In addition to applying adaptive fuzzy algorithms, neural networks techniques and other intelligent algorithms have been deployed to deal with external disturbances. However, all of them are not capable of dealing with complex unknown disturbance, and the system could become trapped into local minima. In order to alleviate the disadvantages of existing works, the observer reconstruction methodology is deployed to obtain real-time external state information so as to achieve effective identification. Numerous observer-based control methods have been proposed to improve the performance of the USV control system. In [25], a nonlinear observer was proposed to recover the position and velocity of a USV from measured data corrupted by noise. In [26], a novel lumped perturbation observer-based robust control method is proposed to improve the control performance of the system and deal with disturbances. In [27], a finite time disturbance observer is proposed to deal with severe model nonlinearities with large parametric uncertainties and external disturbances. At the same time, strict experimental analysis verifies the effectiveness of the designed observer. In [28], a finite-time disturbance observer was proposed to observe fault-mixed unknowns of USV with input saturation. In order to enhance trajectory tracking performance of an asymmetric

underactuated surface vehicle, a finite-time unknown observer was proposed to exactly identify complex unknowns in [29], and remarkable performance has been achieved. On the basis of the aforementioned disturbance observers, in order to further improve the convergence speed and eliminate the influence of initial observation error on the accuracy of disturbance observer, we combine fixed-time control theory with disturbance observer technology to design a novel accurate disturbance observer (ADO) in this paper, which realizes accurate observation of complex disturbances.

In this paper, in order to solve the formation control problem of USVs with internal unmodeled dynamics and external unknown disturbances, a fixed-time formation control strategy is proposed, which contains uncertain and external disturbances to improve the convergence rate and handling of disturbances. In order to promote the following development, firstly, for the tracking control subsystem, a novel fixed-time fast terminal sliding mode tracking control (FTFTSM-TC) strategy is proposed. Secondly, an internal model of uncertainties and external disturbances in the formation control subsystem is considered, and a novel accurate disturbance observer (ADO) is designed to accurately identify disturbances in the USV formation system. Then, to accurately observe lumped disturbance terms and maintain the desired formation, an ADO-based fixed-time formation control strategy (ADO-FTFC) is proposed. Finally, vigorous analysis using the Lyapunov function shows that the proposed formation control strategy is accurate and reliable. The main contributions of this paper are as follows:

1. Aiming at solving the formation control problem of USVs under complex disturbances, the overall formation control framework is divided into tracking and formation control subsystems. Then, we design the FTFTSM-TC strategy and ADO-FTFC strategy. On the basis of simplifying the formation control structure, the convergence rate and control accuracy of the system are greatly improved by the proposed method, and the convergence rate of the system is shown to be independent of the initial state of the system (Section 3.1).
2. In order to improve the disturbance observation accuracy of the formation control system, we design an ADO to achieve real-time control of disturbances and perform accurate observation of the lumped uncertainty item efficiently in the formation control system (Section 3.2).

The paper is organized as follows: Section 2 introduces preliminaries and the problem formulation of the USV formation system. Section 3 presents the design process of the formation control system and establishes the stability of the entire formation system. The design process of the ADO is also introduced. In Section 4, simulation results which demonstrate that the proposed formation controller and disturbance observer can achieve excellent results are presented. Section 5 concludes the paper.

## 2. Preliminaries and Problem Formulation

This section introduces some key lemmas which are necessary for developing the formation control strategy.

### 2.1. Preliminaries

**Lemma 1.** *Consider the following nonlinear system [30]:*

$$\dot{x}(t) = f(x(t))$$
$$x(0) = 0, f(0) = 0, x \in R^n \tag{1}$$

*where $f(x(t))$ is a nonlinear function defined in the origin neighborhood, and $x = [x_1, x_2, \cdots, x_n]^{\mathrm{T}}$ is the vector of system state. If system (1) has a negative homogeneity degree and is asymptotically stable, the system is finite-time stable.*

**Lemma 2.** *Consider the following scalar system [31]:*

$$\dot{y} = -\gamma_1 y^{2-p/q} - \gamma_2 y^{p/q}, y(0) = y_0 \tag{2}$$

*where $\gamma_1, \gamma_2 > 0$ and $p < q$ and $p, q$ are both positive odd integers. System (2) is fixed-time stable, and the upper bound of the convergence time satisfies:*

$$T_{max}(y_0) = \frac{q\pi}{2\sqrt{\gamma_1\gamma_2}(q-p)}. \tag{3}$$

**Lemma 3.** *Consider the following system [32]:*

$$\dot{y} = -l_a sig^{m_1}(y) - l_b sig^{m_2}(y), y(0) = y_0 \tag{4}$$

*where $m_1 > 1$, $0 < m_2 < 1$ and $l_a, l_b > 0$. When the system equilibrium point is fixed-time stable, the upper bound of the convergence time can be calculated independently of the initial state and is given by:*

$$T_{max} = \frac{1}{l_a(m_1-1)} + \frac{1}{l_b(1-m_2)}. \tag{5}$$

**Lemma 4.** *If there is a continuous radial bounded function, $V : R^n \rightarrow R_+ \cup \{0\}$ satisfies: $V(x) = 0 \Leftrightarrow x = 0$. For any $x(t)$ satisfying $\dot{V}(x) \leqslant -l_1 V(x)^{\sigma_1} - l_2 V(x)^{\sigma_2}$, $l_1, l_2, \sigma_1, \sigma_2$ are positive numbers and $0 < \sigma_1 < 1$, $\sigma_2 > 1$. Then, the system can converge to zero in a fixed time and the convergence time $T$ satisfies [25]:*

$$T \leqslant T_{max} = \frac{1}{l_1(1-\sigma_1)} + \frac{1}{l_2(\sigma_2-1)}. \tag{6}$$

*2.2. Problem Formulation*

USVs have the characteristics of strong coupling, strong nonlinearity, and high complexity in actual maritime navigation. Unmodeled dynamics and various external disturbances cannot be ignored when establishing a USV model. An overly simple USV model lacks practicality and generality, but an overly complicated model will hinder subsequent controller design. As shown in Figure 1, the leader–follower formation control framework is adopted. Based on the distances between the three USVs, a three-degrees-of-freedom model is used to solve the USV formation problem, which revolves around establishing the relationship between the surge speed $u$, sway speed $v$ and yaw angular speed $r$ in this paper.

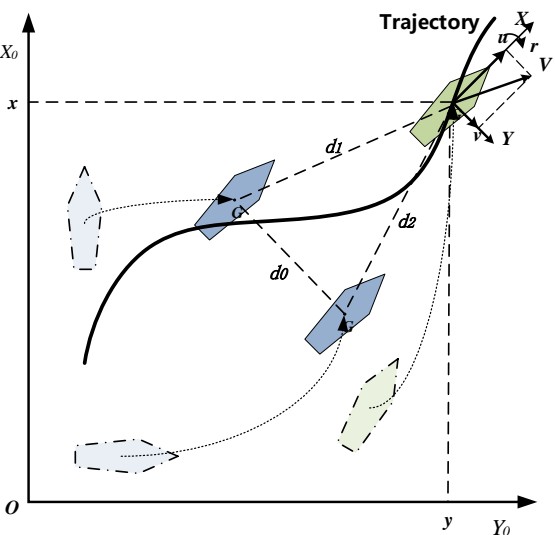

**Figure 1.** Earth-fixed and body-fixed coordinate frames of USVs.

Using the earth-fixed and body-fixed coordinate frames of USVs, the kinetic and dynamic models of the USV can be expressed as follows:

$$\begin{cases} \dot{\eta} = R(\psi)v \\ M\dot{v} + C(v)v + D(v)v = \tau + \delta \end{cases} \tag{7}$$

where $v = [u, v, r]^{\mathrm{T}}$ denotes the velocity vector in the earth-fixed coordinate system, $\eta = [x, y, \psi]^{\mathrm{T}}$ denotes the position and heading angle in the earth-fixed coordinate system, and $\tau = [\tau_{i1}, \tau_{i2}, \tau_{i3}]^{\mathrm{T}}$ denotes the USV control input vector. $\delta = MR^{\mathrm{T}}(\psi)d(t)$ denotes external disturbances caused by wind, waves, and currents; $M = M^{\mathrm{T}} > 0$ is the inertia matrix; $R(\psi)$ is the rotation matrix; $D(v)$ is the damping matrix; and $C(v) = -C(v)^{\mathrm{T}}$ is the skew-symmetric matrix given by:

$$M = \begin{bmatrix} m_{11} & 0 & 0 \\ 0 & m_{22} & m_{23} \\ 0 & m_{32} & m_{33} \end{bmatrix} \tag{8}$$

$$R(\psi) = \begin{bmatrix} \cos\psi & -\sin\psi & 0 \\ \sin\psi & \cos\psi & 0 \\ 0 & 0 & 1 \end{bmatrix} \tag{9}$$

$$D(v) = \begin{bmatrix} d_{11}(v) & 0 & 0 \\ 0 & d_{22}(v) & d_{23}(v) \\ 0 & -d_{32}(v) & d_{33}(v) \end{bmatrix} \tag{10}$$

$$C(v) = \begin{bmatrix} 0 & 0 & c_{13}(v) \\ 0 & 0 & c_{23}(v) \\ -c_{13}(v) & -c_{23}(v) & 0 \end{bmatrix} \tag{11}$$

Note that the aforementioned matrices satisfy the following properties:

$$\begin{aligned} \dot{R}(\psi) &= R(\psi)S(r) \\ R^{\mathrm{T}}(\psi)S(r)R(\psi) &= R(\psi)S(r)R^{\mathrm{T}}(\psi) = S(r) \\ \|R(\psi)\| &= 1, \ R^{\mathrm{T}}(\psi)R(\psi) = \mathrm{I}, \ \forall \psi \in [0, 2\pi] \end{aligned} \tag{12}$$

The $S(r)$ matrix is given by:

$$S(r) = \begin{bmatrix} 0 & -r & 0 \\ r & 0 & 0 \\ 0 & 0 & 0 \end{bmatrix} \tag{13}$$

Rewriting the USV mathematical model to the following Lagrangian mathematical model, we have:

$$M'(\eta)\ddot{\eta} + C'(\eta, \dot{\eta})\dot{\eta} + D'(\eta, \dot{\eta})\dot{\eta} = R(\eta)\tau + \delta(t) \tag{14}$$

where $M'$, $C'$ and $D'$ can be expressed as follows:

$$\begin{aligned} M'(\eta) &= R(\eta)MR^{\mathrm{T}}(\eta) \\ C'(\eta, \dot{\eta}) &= R(\eta)(C - MS)R^{\mathrm{T}}(\eta) \\ D'(\eta, \dot{\eta}) &= R(\eta)DR^{\mathrm{T}}(\eta) \\ M(\eta) &= M^{\mathrm{T}}(\eta) > 0 \\ M(\eta) - 2C(\eta, \dot{\eta}) &= -[M(\eta) - 2C(\eta, \dot{\eta})]^{\mathrm{T}} \end{aligned} \tag{15}$$

To facilitate the following analysis, we define:

$$\begin{cases} \dot{x}_1 = x_2 \\ \dot{x}_2 = M'^{-1}(x_1)R(x_1)\tau + Z(\cdot) \end{cases} \tag{16}$$

where $\mathbf{Z}(\cdot) = \boldsymbol{M'}^{-1}[\boldsymbol{\delta}(t) - \boldsymbol{C}^*(\boldsymbol{x_1}, \boldsymbol{x_2})\boldsymbol{x_2} - \boldsymbol{D}^*(\boldsymbol{x_1}, \boldsymbol{x_2})\boldsymbol{x_2}]$, $\boldsymbol{x_1} = \boldsymbol{\eta}$, $\boldsymbol{x_2} = \dot{\boldsymbol{\eta}}$, the term $f$ represents the lumped uncertainty term of disturbances in the formation system, which satisfies continuous differentiability and boundedness conditions, i.e., $\|\mathbf{Z}(\cdot)\| \leq \Omega$, where $\Omega$ is a bounded positive constant.

Definitions of each parameter are shown in Table 1. The term $X_*, Y_*, Z_*$ denotes the hydrodynamic derivatives, $I_z$ is the moment of inertia, $m$ is the mass of the USV, and $N_{\dot{v}} = Y_{\dot{r}}$.

**Table 1.** Definitions of parameters in $M, C, D$.

| Parameter | Value | Parameter | Value |
|---|---|---|---|
| $m_{11}$ | $m - X_{\dot{u}}$ | $c_{23}(v)$ | $m_{11}u$ |
| $m_{22}$ | $m - Y_{\dot{v}}$ | $d_{11}(v)$ | $-X_u - X_{\|u\|u}\|u\| - X_{uuu}u^2$ |
| $m_{23}$ | $mx_g - Y_{\dot{r}}$ | $d_{22}(v)$ | $-Y_v - Y_{\|v\|v}\|v\|$ |
| $m_{32}$ | $mx_g - N_{\dot{v}}$ | $d_{23}(v)$ | $-Y_r - Y_{\|v\|r}\|v\| - Y_{\|r\|r}\|r\|$ |
| $m_{33}$ | $I_z - N_{\dot{r}}$ | $d_{32}(v)$ | $-N_v - N_{\|v\|v}\|v\| - N_{\|r\|v}\|r\|$ |
| $c_{13}(v)$ | $m_{11} - m_{23}r$ | $d_{33}(v)$ | $-N_r - N_{\|v\|r}\|v\| - N_{\|r\|r}\|r\|$ |

The desired trajectory is given by:

$$\begin{cases} \dot{\boldsymbol{\eta}}_d = \boldsymbol{R}(\psi_d)\boldsymbol{v}_d \\ \boldsymbol{M}'_d \dot{\boldsymbol{v}}_d + \boldsymbol{C}(\boldsymbol{v}_d)\boldsymbol{v}_d + \boldsymbol{D}(\boldsymbol{v}_d)\boldsymbol{v}_d = \boldsymbol{\tau}_d \end{cases} \tag{17}$$

where $\boldsymbol{\eta} = [x_d, y_d, \psi_d]^T$ is the desired position vector, $\boldsymbol{v}_d = [u_d, v_d, r_d]^T$ is the velocity vector of the USV, and $\boldsymbol{\tau}_d = [\tau_{d1}, \tau_{d2}, \tau_{d3}]^T$ is the desired control input.

## 3. Design of the Proposed Controller

As shown in Figure 2, the formation system is divided into a tracking control subsystem and a formation control subsystem to facilitate controller design. Moreover, for ease of analysis, lumped disturbances are not considered when the trajectory tracking controller is designed.

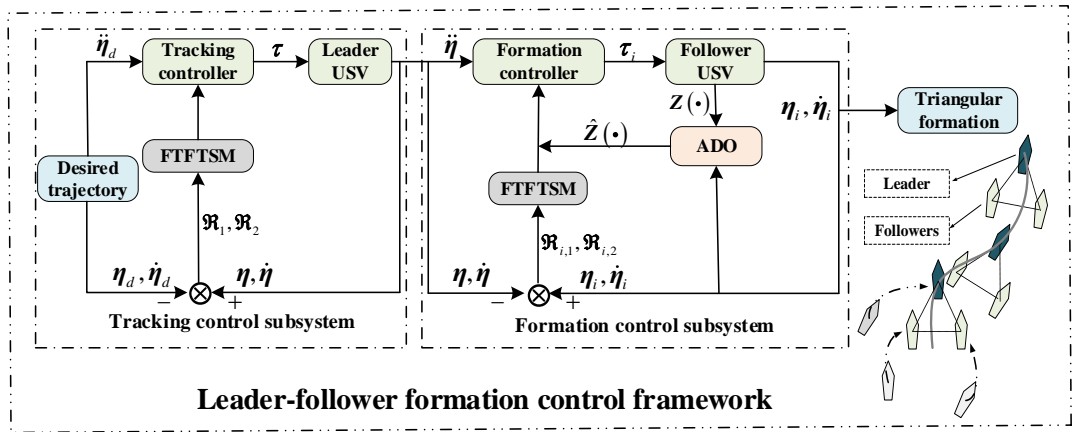

**Figure 2.** The algorithm architecture of the integrated USV formation system.

### 3.1. Tracking Control Subsystem

Dynamic tracking errors of the leader USV and the desired trajectory are defined as follows:

$$\begin{cases} \boldsymbol{\Re_1} = \boldsymbol{x_1} - \boldsymbol{\eta}_d \\ \boldsymbol{\Re_2} = \boldsymbol{x_2} - \dot{\boldsymbol{\eta}}_d \end{cases} \tag{18}$$

It follows from (16) and (18) that:

$$\begin{cases} \dot{\mathfrak{R}}_1 = \mathfrak{R}_2 \\ \dot{\mathfrak{R}}_2 = M'^{-1}(x_1)R(x_1)\tau + Z(\cdot) - \ddot{\eta}_d \end{cases} \tag{19}$$

A novel fixed-time fast terminal sliding mode (FTFTSM) is designed to ensure that the USV formation control system has a faster convergence rate and converge in fixed time in the entire universe, shown as follows:

$$s = \mathfrak{R}_1 + \left[ \frac{\mathfrak{R}_2}{a_1 \mathfrak{R}_1^{H-N} + b_1} \right]^{\frac{1}{N}} \tag{20}$$

The derivation is as follows:

$$\dot{s} = \mathfrak{R}_2 + \frac{1}{N} \left[ \left( a_1 \mathfrak{R}_1^{H-N} + b_1 \right)^{-1} \mathfrak{R}_2 \right]^{\frac{1}{N}-1} \\ \left[ -a_1(H-N)(a_1 \mathfrak{R}_1^{H-N} + b_1)^{-2} \mathfrak{R}_1^{H-N-1} \mathfrak{R}_2^2 + (a_1 \mathfrak{R}_1^{H-N} + b_1)^{-1} \dot{\mathfrak{R}}_2 \right] \tag{21}$$

where $a_1 > 0$, $b_1 > 0$, $H = \frac{h_1}{h_2}$, $N = \frac{n_1}{n_2}$, $h_1, h_2, n_1, n_2$ are positive odd numbers and satisfy $h_1 > h_2$, $n_1 < n_2$, $H - N > 1$.

In order to simplify the controller design, let $\mathfrak{S} = (a_1 \mathfrak{R}_1^{H-N} + b_1)^{-1}$, simplified as follows:

$$\dot{s} = \mathfrak{R}_2 + \frac{1}{N} (\mathfrak{S}\mathfrak{R}_2)^{\frac{1}{N}-1} [-a_1(H-N)\mathfrak{S}^2 \mathfrak{R}_1^{H-N-1} \mathfrak{R}_2^2 + \mathfrak{S}\dot{\mathfrak{R}}_2] \\ = \mathfrak{R}_2 - \frac{a_1(H-N)}{N} \mathfrak{S}^{\frac{1}{N}+1} \mathfrak{R}_1^{H-N-1} \mathfrak{R}_2^{\frac{1}{N}+1} + \frac{1}{N} \mathfrak{S}^{\frac{1}{N}} \mathfrak{R}_2^{\frac{1}{N}-1} \dot{\mathfrak{R}}_2 \tag{22}$$

Combined with Lemma 2, the FTFTSM-TC strategy is designed as follows:

$$\tau = -M(x_1)'R(x_1)^{-1} [\mathfrak{R}_2 - \frac{N}{\mathfrak{S}^{\frac{1}{N}} \mathfrak{R}_2^{\frac{1}{N}-1}} (\frac{a_1(H-N)}{N} \mathfrak{S}^{\frac{1}{N}+1} \mathfrak{R}_1^{H-N-1} \mathfrak{R}_2^{\frac{1}{N}+1} \\ + \varepsilon_0 s + \varepsilon_1 s^{2-p/q} + \varepsilon_2 s^{p/q}) - \ddot{\eta}_d] \tag{23}$$

where $\varepsilon_0, \varepsilon_1, \varepsilon_2$ are the control coefficients of the controller, and $p, q$ are positive odd numbers satisfying $p < q$.

Now, the main results of this work are presented as follows.

**Theorem 1.** *Consider the USV tracking control system governed by (19) under the assumption that there is no lumped disturbance, the proposed FTFTSM-TC strategy can ensure that velocity vector $\dot{\eta}$ and the position vector $\eta$ of the leader USV can accurately track the desired trajectory in fixed time.*

**Proof of Theorem 1.** Reaching phase: Here, we establish that the errors $\mathfrak{R}_1, \mathfrak{R}_2$ can reach the sliding surface in fixed time.

The following Lyapunov function is selected:

$$V = \frac{1}{2} s^{\mathrm{T}} s \tag{24}$$

The derivation is as follows:

$$\begin{aligned}
\dot{V} &= \mathbf{s}^{\mathrm{T}}\dot{\mathbf{s}} \\
&= \mathbf{s}^{\mathrm{T}}[\mathfrak{R}_2 - \tfrac{a_1(H-N)}{N}\mathfrak{S}^{\frac{1}{N}+1}\mathfrak{R}_1^{H-N-1}\mathfrak{R}_2^{\frac{1}{N}+1} + \tfrac{1}{N}\mathfrak{S}^{\frac{1}{N}}\mathfrak{R}_2^{\frac{1}{N}-1}\dot{\mathfrak{R}}_2] \\
&= \mathbf{s}^{\mathrm{T}}[\mathfrak{R}_2 - \tfrac{a_1(H-N)}{N}\mathfrak{S}^{\frac{1}{N}+1}\mathfrak{R}_1^{H-N-1}\mathfrak{R}_2^{\frac{1}{N}+1} \\
&\quad + \tfrac{1}{N}\mathfrak{S}^{\frac{1}{N}}\mathfrak{R}_2^{\frac{1}{N}-1}(M'^{-1}(x_1)R(x_1)\boldsymbol{\tau} - \ddot{\boldsymbol{\eta}}_d)] \\
&= -\mathbf{s}^{\mathrm{T}}[\varepsilon_0\mathbf{s} + \varepsilon_1\mathbf{s}^{2-p/q} + \varepsilon_2\mathbf{s}^{p/q}] \\
&= -\varepsilon_0\mathbf{s}^2 - \varepsilon_1 2^{\frac{3q-p}{2q}}(\tfrac{1}{2}\mathbf{s}^2)^{\frac{3q-p}{2q}} - \varepsilon_2 2^{\frac{q+p}{2q}}(\tfrac{1}{2}\mathbf{s}^2)^{\frac{q+p}{2q}} \\
&\leqslant -\varepsilon_1 2^{\frac{3q-p}{2q}}\mathrm{V}^{2-\frac{p+q}{2q}} - \varepsilon_2 2^{\frac{p+q}{2q}}\mathrm{V}^{\frac{p+q}{2q}}
\end{aligned} \tag{25}$$

According to Lemma 2, the upper bound of the convergence time is computed by:

$$T_0 = \frac{q\pi}{2\sqrt{\varepsilon_1\varepsilon_2}(q-p)} \tag{26}$$

Then, in the maximum set time $T_0$, the sliding surface $\mathbf{s}$ can be reached. After the sliding surface arrives, $\mathbf{s} = 0, \dot{\mathbf{s}} = 0$, and we have:

$$\mathfrak{R}_2 = \dot{\mathfrak{R}}_1 = -a_1\mathfrak{R}_1^H - b_1\mathfrak{R}_1^N \tag{27}$$

Furthermore, according to Lemma 3, the tracking errors $\mathfrak{R}_1$, $\mathfrak{R}_2$ will converge to zero along the manifold in a fixed time. In summary, the controller designed for the tracking subsystem can guarantee that $\boldsymbol{\eta} = \boldsymbol{\eta}_d, \boldsymbol{v} = \boldsymbol{v}_d$ in fixed time, i.e., the leader USV can track the desired trajectory accurately.

Theorem 1 is proven complete. □

### 3.2. Formation Control Subsystem

We consider the lumped disturbance $\mathbf{Z}(\cdot)$ in the design process of the formation control subsystem, which includes the external environment disturbance $\delta$ and the internal disturbance. The internal disturbance contains items $C^*(x_{i,1}, x_{i,2})x_{i,2}$ and $D^*(x_{i,1}, x_{i,2})x_{i,2}$ related to the internal unmodeled dynamics of the USV. Definition $C^\dagger(x_{i,1}, x_{i,2})x_{i,2} = C(x_{i,1}, x_{i,2})x_{i,2} - C^*(x_{i,1}, x_{i,2})x_{i,2}$ and $D^\dagger(x_{i,1}, x_{i,2})x_{i,2} = D(x_{i,1}, x_{i,2})x_{i,2} - D^*(x_{i,1}, x_{i,2})x_{i,2}$.

First, we rewrite the Lagrangian model of the follower USVs:

$$\begin{cases} \dot{x}_{i,1} = x_{i,2} \\ \dot{x}_{i,2} = M'^{-1}(x_{i,1})[\boldsymbol{\tau}_i - C^\dagger(x_{i,1}, x_{i,2})x_{i,2} - D^\dagger(x_{i,1}, x_{i,2})x_{i,2}] + \mathbf{Z}(\cdot) \end{cases} \tag{28}$$

where $i = 1, 2$ denotes follower USV1 and USV2, and $\mathbf{Z}(\cdot)$ is defined as follows:

$$\mathbf{Z}(\cdot) = M'^{-1}[\delta(t) - C^*(x_{i,1}, x_{i,2})x_{i,2} - D^*(x_{i,1}, x_{i,2})x_{i,2}] \tag{29}$$

where $x_{i,1} = \boldsymbol{\eta}_i$ and $x_{i,2} = \dot{\boldsymbol{\eta}}_i$.

**Assumption 1.** *The term $\mathbf{Z}(\cdot)$ represents external disturbances of the formation control subsystem and the internal lumped uncertainty items, which is satisfying the conditions of boundedness and continuous differentiability, i.e., $\|\mathbf{Z}(\cdot)\| \leq \Omega$, where $\Omega$ is a bounded positive constant.*

Auxiliary variables are defined as follows:

$$\begin{aligned}
\boldsymbol{\Theta} &= x_{i,2} - \boldsymbol{\omega} \\
\dot{\boldsymbol{\omega}} &= M'^{-1}(x_{i,1})[\boldsymbol{\tau}_i - C^\dagger(x_{i,1}, x_{i,2})x_{i,2} - D^\dagger(x_{i,1}, x_{i,2})x_{i,2}] \\
&\quad + \ell_1\boldsymbol{\Theta} + \ell_2\boldsymbol{\Theta}^{\kappa_1} + \ell_3\boldsymbol{\Theta}^{\kappa_2} + \ell_4 sign(\boldsymbol{\Theta})
\end{aligned} \tag{30}$$

where $\ell_1, \ell_2, \ell_3, \ell_4$ are positive definite diagonal matrices satisfying $\ell_4 \geqslant \Omega$. $\kappa_1, \kappa_2$ are positive numbers and satisfy $0 < \kappa_1 < 1, \kappa_2 > 1$.

The derivative of $\boldsymbol{\Theta}$ is as follows:

$$
\begin{aligned}
\dot{\boldsymbol{\Theta}} &= \dot{x}_{i,2} - \dot{\omega} \\
&= \boldsymbol{M}^{'-1}(x_{i,1})[\boldsymbol{\tau}_i - \boldsymbol{C}^\dagger(x_{i,1}, x_{i,2})x_{i,2} - \boldsymbol{D}^\dagger(x_{i,1}, x_{i,2})x_{i,2}] + \boldsymbol{Z}(\cdot) \\
&\quad - \boldsymbol{M}^{'-1}(x_{i,1})[\boldsymbol{\tau}_i - \boldsymbol{C}^\dagger(x_{i,1}, x_{i,2})x_{i,2} - \boldsymbol{D}^\dagger(x_{i,1}, x_{i,2})x_{i,2}] \\
&\quad - [\ell_1\boldsymbol{\Theta} + \ell_2\boldsymbol{\Theta}^{\kappa_1} + \ell_3\boldsymbol{\Theta}^{\kappa_2} + \ell_4 sign(\boldsymbol{\Theta})] \\
&= \boldsymbol{Z}(\cdot) - [\ell_1\boldsymbol{\Theta} + \ell_2\boldsymbol{\Theta}^{\kappa_1} + \ell_3\boldsymbol{\Theta}^{\kappa_2} + \ell_4 sign(\boldsymbol{\Theta})]
\end{aligned}
\tag{31}
$$

The ADO is designed as follows:

$$
\hat{\boldsymbol{Z}}(\cdot) = \ell_1\boldsymbol{\Theta} + \ell_2\boldsymbol{\Theta}^{\kappa_1} + \ell_3\boldsymbol{\Theta}^{\kappa_2} + \ell_4 sign(\boldsymbol{\Theta})
\tag{32}
$$

$\tilde{\boldsymbol{Z}}(\cdot)$ is the observation error, which is defined as follows:

$$
\begin{aligned}
\tilde{\boldsymbol{Z}}(\cdot) &= \hat{\boldsymbol{Z}}(\cdot) - \boldsymbol{Z}(\cdot) \\
&= \ell_1\boldsymbol{\Theta} + \ell_2\boldsymbol{\Theta}^{\kappa_1} + \ell_3\boldsymbol{\Theta}^{\kappa_2} + \ell_4 sign(\boldsymbol{\Theta}) + \\
&\quad \boldsymbol{M}^{'-1}(x_{i,1})[\boldsymbol{\tau}_i - \boldsymbol{C}^\dagger(x_{i,1}, x_{i,2})x_{i,2} - \boldsymbol{D}^\dagger(x_{i,1}, x_{i,2})x_{i,2}] - \dot{x}_{i,2} \\
&= \dot{\omega} - \dot{x}_{i,2} \\
&= -\dot{\boldsymbol{\Theta}}
\end{aligned}
\tag{33}
$$

From the above formula, if $\dot{\boldsymbol{\Theta}}$ converges, then $\tilde{\boldsymbol{Z}}(\cdot)$ converges. We select the following Lyapunov function:

$$
V_Z = \frac{1}{2}\boldsymbol{\Theta}^T\boldsymbol{\Theta}
\tag{34}
$$

The derivation of the above formula is as follows:

$$
\begin{aligned}
\dot{V}_Z &= \boldsymbol{\Theta}^T\dot{\boldsymbol{\Theta}} \\
&= \boldsymbol{\Theta}^T[\boldsymbol{Z}(\cdot) - \ell_1\boldsymbol{\Theta} - \ell_2\boldsymbol{\Theta}^{\kappa_1} - \ell_3\boldsymbol{\Theta}^{\kappa_2} - \ell_4 sign(\boldsymbol{\Theta})] \\
&\leqslant -\ell_1\boldsymbol{\Theta}^T\boldsymbol{\Theta} - \ell_2\boldsymbol{\Theta}^T\boldsymbol{\Theta}^{\kappa_1} - \ell_3\boldsymbol{\Theta}^T\boldsymbol{\Theta}^{\kappa_2} \\
&\leqslant -2^{\frac{\kappa_1+1}{2}}\ell_{2,min}\left(\frac{1}{2}\boldsymbol{\Theta}^T\boldsymbol{\Theta}\right)^{\frac{\kappa_1+1}{2}} - 2^{\frac{\kappa_2+1}{2}}\ell_{3,min}\left(\frac{1}{2}\boldsymbol{\Theta}^T\boldsymbol{\Theta}\right)^{\frac{\kappa_2+1}{2}} \\
&= -\alpha_{z,1}V_Z^{\beta_{z,1}} - -\alpha_{z,2}V_Z^{\beta_{z,2}}
\end{aligned}
\tag{35}
$$

where $\alpha_{z,1} = 2^{\frac{\kappa_1+1}{2}}\ell_{2,min}$, $\alpha_{z,2} = 2^{\frac{\kappa_2+1}{2}}\ell_{3,min}$, $0 < \beta_{z,1} = \frac{\kappa_1+1}{2} < 1$, $\beta_{z,2} = \frac{\kappa_2+1}{2} > 1$.

Furthermore, according to Lemma 4, $\boldsymbol{\Theta}$ is globally fixed time stable. The convergence time $T_z$ is as follows:

$$
T_Z \leqslant T_{Z,max} = \frac{1}{\alpha_{z,1}(1-\beta_{z,1})} + \frac{1}{\alpha_{z,2}(\beta_{z,2}-1)}
\tag{36}
$$

Then we can obtain:

$$
\dot{\boldsymbol{\Theta}} = 0 \rightarrow \tilde{\boldsymbol{Z}}(\cdot) = 0, t \geqslant T_Z
\tag{37}
$$

In summary, the designed ADO can accurately observe the disturbances in fixed time and ensure that the observation error is independent of the initial observation error.

**Theorem 2.** *Consider the USV formation control system with a lumped uncertainty term $\boldsymbol{Z}(\cdot)$ governed by (29): the designed ADO can accurately identify the disturbances. Moreover, the proposed ADO-FTFC strategy can ensure the position vector $\boldsymbol{\eta}_i$, and velocity vector $\dot{\boldsymbol{\eta}}_i$ of the followers can accurately track the velocity vector $\dot{\boldsymbol{\eta}}$ and position vector $\boldsymbol{\eta}$ of the leader and maintain the desired formation in fixed time.*

**Proof of Theorem 2.** The dynamic error between the leader and the follower USV is defined as follows:

$$
\begin{cases}
\Re_{i,1} = x_{i,1} - \eta \\
\Re_{i,2} = x_{i,2} - \dot{\eta}
\end{cases}
\tag{38}
$$

The derivation of the above formula is as follows:

$$
\begin{cases}
\dot{\Re}_{i,1} = \Re_{i,2} \\
\dot{\Re}_{i,2} = M'^{-1}(x_{i,1})R(x_{i,1})\tau_i + Z(\cdot) - \ddot{\eta}
\end{cases}
\tag{39}
$$

Considering the FTFTSM (20), we have:

$$
s_i = \Re_{i,1} + \left[ \frac{\Re_{i,2}}{a_{i,1}\Re_{i,1}^{H_i - N_i} + b_{i,1}} \right]^{\frac{1}{N_i}}
\tag{40}
$$

The derivation is as follows:

$$
\dot{s}_i = \Re_{i,2} - \frac{a_{i,1}(H_i - N_i)}{N_i}\Im_i^{\frac{1}{N_i}+1}\Re_{i,1}^{H_i-N_i-1}\Re_{i,2}^{\frac{1}{N_i}+1} + \frac{1}{N_i}\Im_i^{\frac{1}{N_i}}\Re_{i,2}^{\frac{1}{N_i}-1}\dot{\Re}_{i,2}
\tag{41}
$$

where $i = 1, 2$. We design the ADO-FTFC strategy as follows:

$$
\tau_i = -M(x_{i,1})'R(x_{i,1})^{-1}\Big[\Re_{i,2} - \frac{N_i}{\Im_i^{\frac{1}{N_i}}\Re_{i,2}^{\frac{1}{N_i}-1}}\Big(\frac{a_{i,1}(H_i - N_i)}{N_i}\Im_i^{\frac{1}{N_i}+1}\Re_{i,1}^{H_i-N_i-1}\Re_{i,2}^{\frac{1}{N_i}+1}
$$
$$
+\varepsilon_{i,0}s_i + \varepsilon_{i,1}s_i^{2-p/q} + \varepsilon_{i,2}s_i^{p/q}\Big) + \hat{Z}(\cdot) - \ddot{\eta}\Big]
\tag{42}
$$

where $\varepsilon_{i,0}, \varepsilon_{i,1}, \varepsilon_{i,2}$ are the control coefficients of the controller, and $p, q$ are positive odd numbers satisfying $p < q$.

The following Lyapunov function is selected to prove that the errors $\Re_{i,1}, \Re_{i,2}$ can reach the sliding surface in a fixed time:

$$
V_i = \frac{1}{2}s_i^{\mathrm{T}}s_i
\tag{43}
$$

The derivation is as follows:

$$
\begin{aligned}
\dot{V}_i &= s_i^{\mathrm{T}}\dot{s}_i \\
&= s_i^{\mathrm{T}}\Big[\Re_{i,2} - \frac{a_{i,1}(H_i-N_i)}{N_i}\Im_i^{\frac{1}{N_i}+1}\Re_{i,1}^{H_i-N_i-1}\Re_{i,2}^{\frac{1}{N_i}+1} + \frac{1}{N_i}\Im_i^{\frac{1}{N_i}}\Re_{i,2}^{\frac{1}{N_i}-1}\dot{\Re}_{i,2}\Big] \\
&= s_i^{\mathrm{T}}\Big[\Re_{i,2} - \frac{a_{i,1}(H_i-N_i)}{N_i}\Im_i^{\frac{1}{N_i}+1}\Re_{i,1}^{H_i-N_i-1}\Re_{i,2}^{\frac{1}{N_i}+1} \\
&\quad + \frac{1}{N_i}\Im_i^{\frac{1}{N_i}}\Re_{i,2}^{\frac{1}{N_i}-1}(M'^{-1}(x_{i,1})R(x_{i,1})\tau_i + Z(\cdot) - \ddot{\eta}_d)\Big] \\
&= -s_i^{\mathrm{T}}\Big[\varepsilon_{i,0}s_i + \varepsilon_{i,1}s_i^{2-p/q} + \varepsilon_{i,2}s_i^{p/q} + (Z(\cdot) - \hat{Z}(\cdot))\Big] \\
&= -s_i^{\mathrm{T}}\Big[\varepsilon_{i,0}s_i + \varepsilon_{i,1}s_i^{2-p/q} + \varepsilon_{i,2}s_i^{p/q}\Big] \\
&= -\varepsilon_{i,0}s_i^2 - \varepsilon_{i,1}2^{\frac{3q-p}{2q}}\Big(\frac{1}{2}s_i^2\Big)^{\frac{3q-p}{2q}} - \varepsilon_{i,2}2^{\frac{q+p}{2q}}\Big(\frac{1}{2}s_i^2\Big)^{\frac{q+p}{2q}} \\
&\leqslant -\varepsilon_{i,1}2^{\frac{3q-p}{2q}}V_i^{2-\frac{p+q}{2q}} - \varepsilon_{i,2}2^{\frac{p+q}{2q}}V_i^{\frac{p+q}{2q}}
\end{aligned}
\tag{44}
$$

The upper bound of the convergence time is computed according to Lemma 2, shown as follows:

$$
T_i = \frac{q\pi}{2\sqrt{\varepsilon_{i,1}\varepsilon_{i,2}}(q - p)}
\tag{45}
$$

Then, the sliding surface $s_i$ can be reached within the maximum set time $T_0$. After the sliding surface arrives, $s_i = 0, \dot{s}_i = 0$, and we have:

$$
\Re_{i,2} = \dot{\Re}_{i,1} = -a_{i,1}\Re_{i,1}^{H_i} - b_{i,1}\Re_{i,1}^{N_i}
\tag{46}
$$

Furthermore, according to Lemma 3, the formation errors $\mathfrak{R}_{i,1}$, $\mathfrak{R}_{i,2}$ will converge to zero along the manifold in a fixed time, and three USVs can quickly and accurately form and maintain the desired formation.

Theorem 2 is proven complete. □

## 4. Simulation and Discussion

The parameters of the FTFTSM and ADO are shown in Table 2. In order to verify the effectiveness of the FTFTSM-TC strategy and the ADO-FTTC strategy proposed in this paper, the benchmark USV model Cybership II is used. The main parameters of the Cybership II are shown in Table 3. The initial values of the formation system are as follows: $\eta_d(0) = [0, 0, \pi/2]^T$, $\nu_d(0) = [0.2, 0, 0]^T$. The initial states of the leader USV and the follower USV1 and USV2 are as follows: $\eta_0(0) = [-5, 0.5, 0]^T$, $\eta_1(0) = [-5, -2, 0]^T$, $\eta_2(0) = [-5, 2.5, 0]^T$, $\nu_0(0) = \nu_1(0) = \nu_2(0) = [0, 0, 0]^T$.

**Table 2.** Parameter values.

| Parameters | Values | Parameters | Values | Parameters | Values |
|---|---|---|---|---|---|
| $h_1, h_{i,1}$ | 9 | $h_2, h_{i,2}$ | 5 | $n_1, n_{i,1}$ | 3 |
| $n_2, n_{i,2}$ | 5 | $a_1, a_{i,1}$ | 1 | $b_1, b_{i,1}$ | 1 |
| $\varepsilon_0; \varepsilon_{i,0}$ | 0.1; 0.1 | $\varepsilon_1; \varepsilon_{i,1}$ | 0.8; 0.6 | $\varepsilon_3; \varepsilon_{i,3}$ | 0.7; 0.4 |
| $p$ | 7 | $q$ | 9 | $\ell_1$ | $diag[3, 3, 3]$ |
| $\ell_2$ | $diag[7/9, 7/9, 7/9]$ | $\ell_3$ | $diag[5/7, 5/7, 5/7]$ | $\ell_4$ | $diag[8, 7, 8]$ |
| $\kappa_1$ | 7/9 | $\kappa_2$ | 7/5 | | |

**Table 3.** Main parameters of CyberShip II.

| Parameters | Values | Parameters | Values | Parameters | Values |
|---|---|---|---|---|---|
| $m$ | 23.8000 | $Y_v$ | −0.8612 | $X_{\dot\mu}$ | −2.0 |
| $I_z$ | 1.7600 | $Y_{|v|v}$ | −36.2823 | $Y_{\dot v}$ | −10.0 |
| $x_g$ | 0.460 | $Y_r$ | 0.1079 | $Y_{\dot r}$ | 0.0 |
| $X_\mu$ | −0.7225 | $N_v$ | 0.1052 | $N_{\dot v}$ | 0.0 |
| $X_{|\mu|\mu}$ | −1.3274 | $N_{|v|v}$ | 5.0437 | $N_{\dot r}$ | −1.0 |
| $X_{\mu\mu\mu}$ | −5.8664 | | | | |

The following disturbances are used in the simulation study:

$$d_i = \begin{bmatrix} 5\cos(\frac{\pi}{10}t - \frac{\pi}{3}) \\ 4\cos(\frac{\pi}{5}t + \frac{\pi}{2}) \\ 3\cos(\frac{\pi}{10}t + \frac{\pi}{6}) \end{bmatrix} \tag{47}$$

Simulation results are shown in Figures 3–10. Figure 3 shows the comparison results of the proposed FTFTSM-TC strategy with the ISM-TC strategy. The desired trajectory is set as follows: $\eta_d = [3\sin(0.04t) + 0.2, -1.5\sin(0.06t), \frac{\pi}{2}\cos(0.03t)]^T$. The results show that the ISM-TC strategy cannot deal with external disturbances in real time, and the FTFTSM-TC strategy proposed can achieve accurate tracking control. The FTFTSM-TC strategy proposed can ensure that the leader USV accurately track the desired trajectory and maintain stable movement along the desired trajectory.

In order to compare position and velocity tracking performances in different directions, Figures 4 and 5 show position tracking and velocity tracking under the FTFTSM-TC strategy and the ISM-TC strategy. They clearly demonstrate that the FTFTSM-TC strategy is superior to the ISM-TC strategy.

Figure 6 shows the performance of the ADO disturbance observation. Simulation results show that the ADO strategy proposed can quickly and accurately handle the lumped uncertainty item of the formation system. The norm and derivatives of the tracking errors are shown in Figure 7, which clearly show that the approach of handling the lumped disturbances by the proposed ADO is effective and efficient.

Figure 8 demonstrates that all the USVs maintain a stable straight triangular formation operation, which show that three USVs quickly form the desired formation from different positions in five seconds and maintain stable kinestate and verify closed-loop stability and the effectiveness of the proposed ADO-FTFC strategy. In order to compare control performances on different positions and velocities, Figures 9 and 10 show velocity tracking error and the position tracking error when the leader USV and two follower USVs maintain triangular formation. The results clearly demonstrate that the ADO-FTFC strategy proposed can ensure that the leader and follower USVs can maintain accurate and stable formation. From the change curve of control input with time shown in Figures 11 and 12, it is verified that the designed controller can be stable in a fixed time.

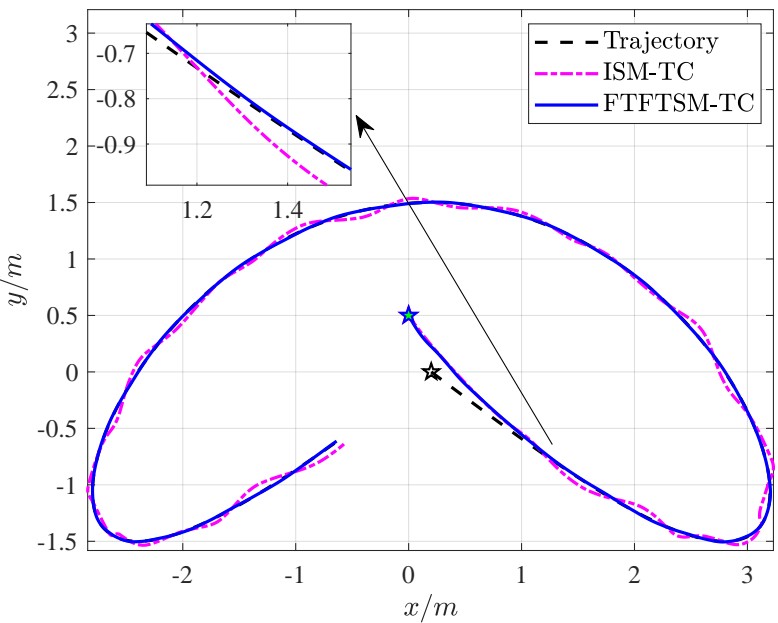

**Figure 3.** Tracking curve of FTFTSM-TC and ISM-TC. The asterisk represents the start position of the USV.

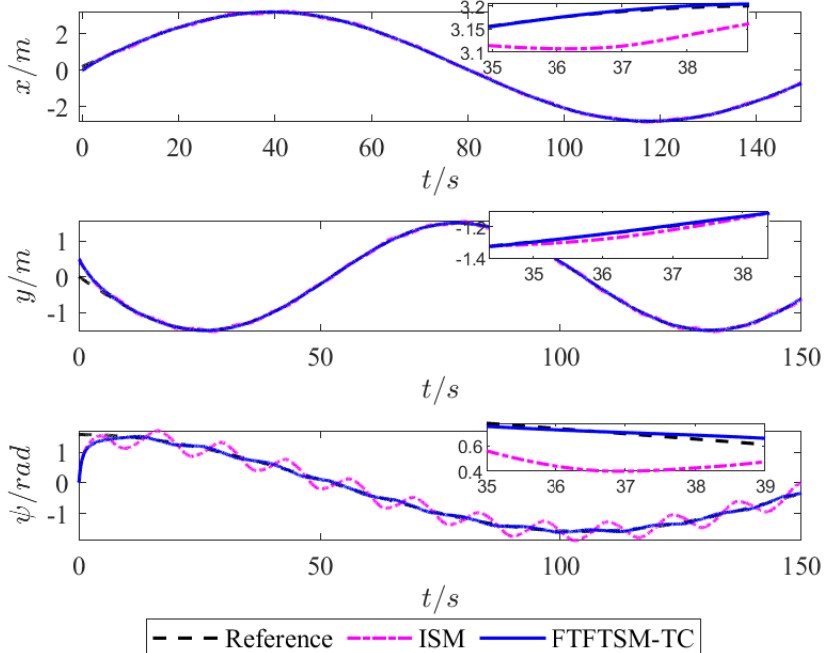

**Figure 4.** Position tracking of FTFTSM-TC and ISM-TC.

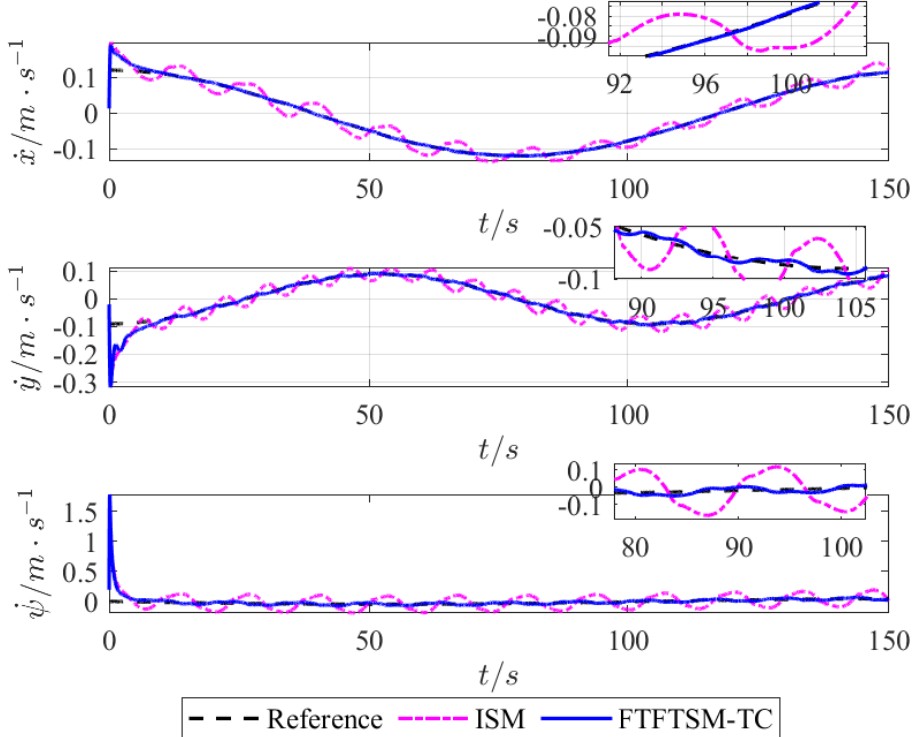

**Figure 5.** Velocity tracking of FTFTSM-TC and ISM-TC.

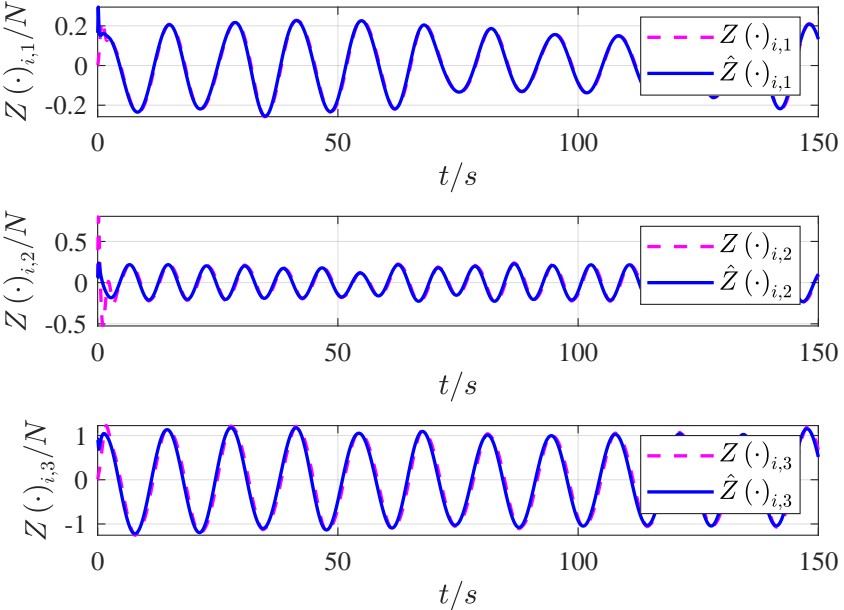

**Figure 6.** Observation results of ADO.

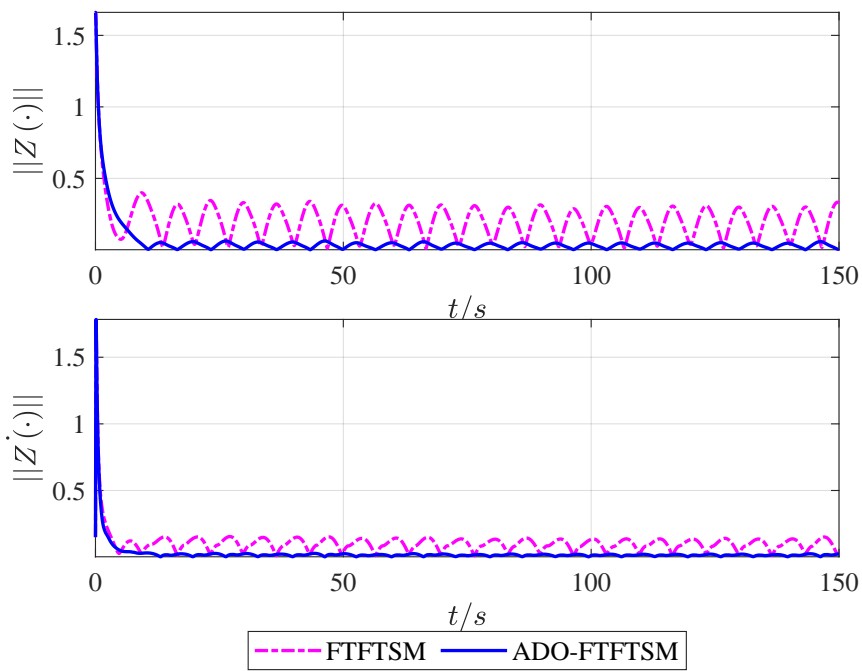

**Figure 7.** Norm of tracking errors and their derivatives.

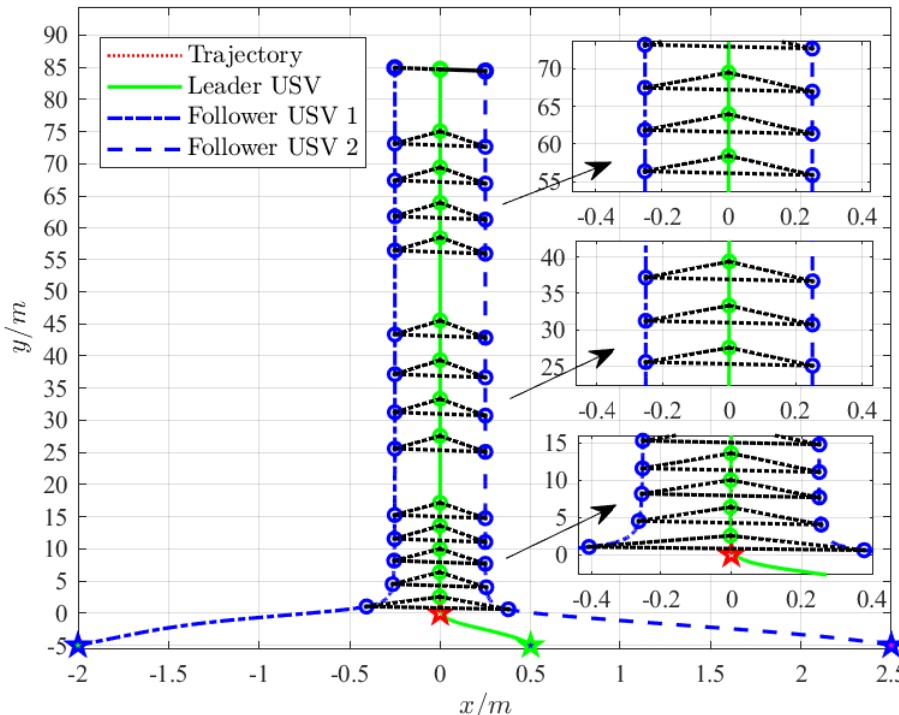

**Figure 8.** Triangular formation of USVs. The asterisk represents the start position of the USV.

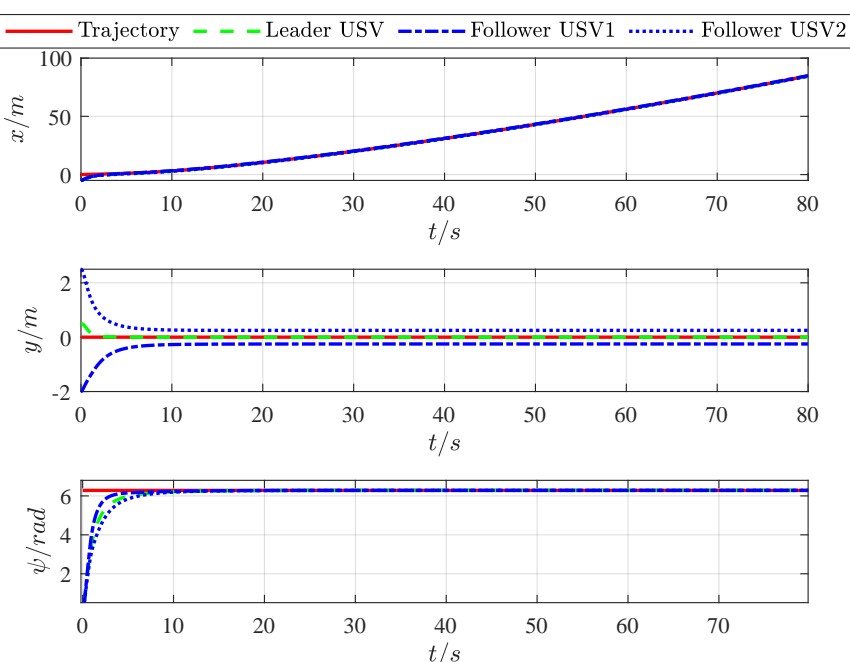

**Figure 9.** Position tracking of triangular formation of USVs.

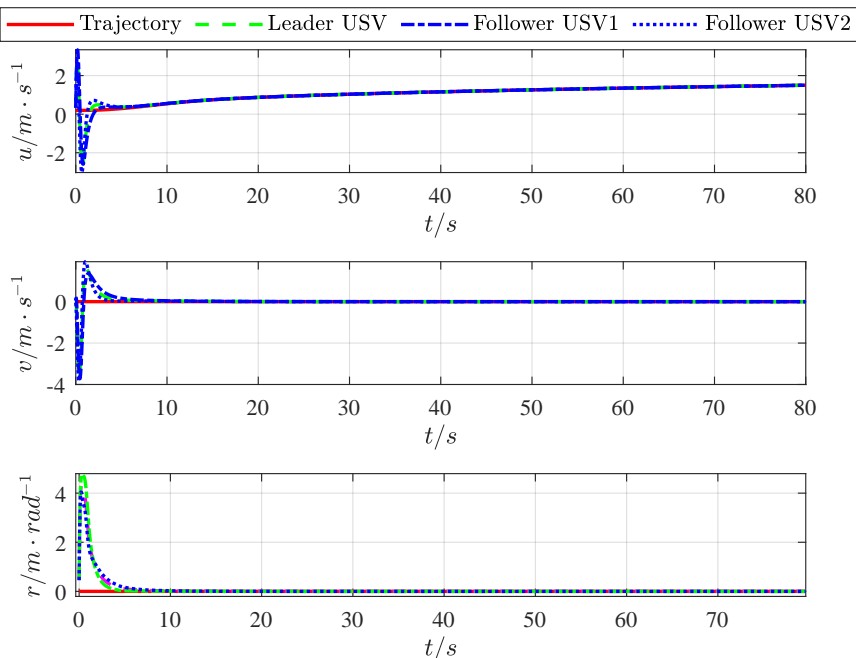

**Figure 10.** Velocity tracking of triangular formation of USVs.

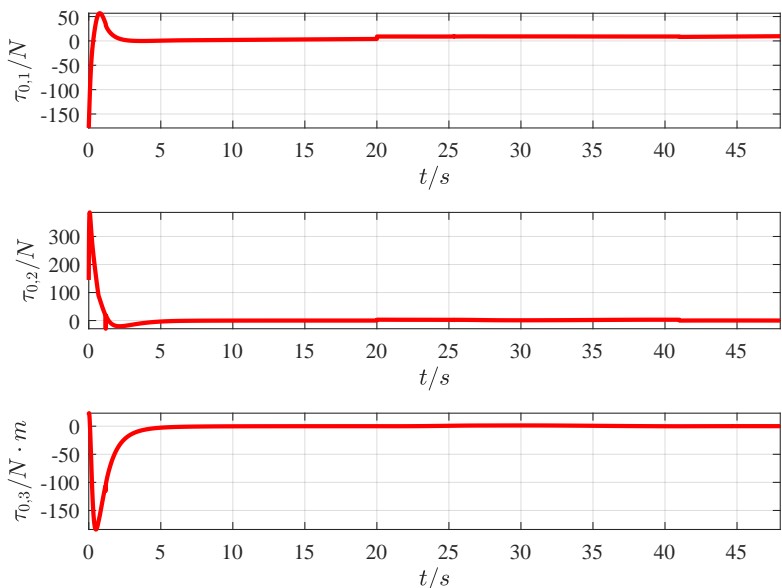

**Figure 11.** Position tracking of triangular formation of USVs.

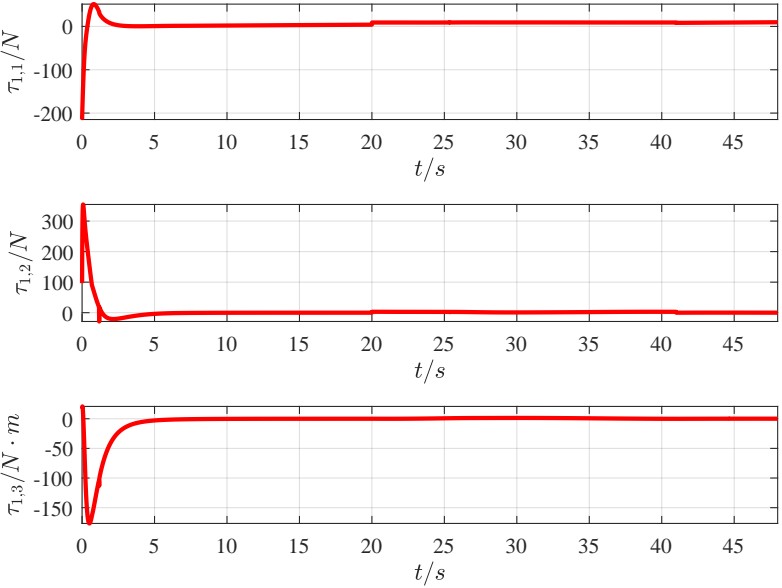

**Figure 12.** Velocity tracking of triangular formation of USVs.

## 5. Conclusions

In order to solve the key problems in the field of cooperative control of USVs, the leader–follower formation control strategy of USVs under unknown disturbances has been successfully designed. We divide the entire formation control system into the tracking control subsystem and the formation control subsystem. In the tracking control subsystem, the FTFTSM-TC strategy is proposed to improve the convergence rate and precision of the tracking control system. In the formation control subsystem, the ADO-FTFC strategy is proposed to observe lumped disturbances, and excellent disturbance identification results have been achieved, thereby ensuring stable and effective USV formation control. Rigorous stability analysis and simulation studies demonstrate that the proposed strategy is superior to the state-of-the-art methods. However, there are still some limitations in this paper. We design the controller based on the leader follower formation framework, which simplifies the interaction between individuals but reduces the flexibility of the formation. Therefore, this is the problem we need to solve in the future. The research results of this paper provide a bright research direction for the future USV formation control.

**Author Contributions:** Conceptualization, Y.Y.; methodology, Y.Y.; formal analysis, H.S. and X.Q.; software, X.Q. and H.S.; data curation, X.Q.; resources, H.S.; writing—original draft preparation, H.S.; writing—review and editing, Y.Y. and X.Q.; project administration, Y.Y.; funding acquisition, H.S. and Y.Y. All authors have read and agreed to the published version of the manuscript.

**Funding:** This research was partially supported by: (a) the project of Intelligent Ship Testing and Verification, (2018/473); (b) the Fundamental Research Funds for the Central Universities (no. 3132019011); (c) the Natural Science Foundation of Liaoning Province (no. 2021-MS-141).

**Institutional Review Board Statement:** Not applicable.

**Informed Consent Statement:** Not applicable.

**Data Availability Statement:** Partial data can be found in this paper.

**Conflicts of Interest:** The authors declare that there are no conflict of interest in the publication of this paper.

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
