# Peer review of "Fixed-Time Formation Control for Unmanned Surface Vehicles with Parametric Uncertainties and Complex Disturbance"

_jmse, doi:10.3390/jmse10091246_

Round 1

Reviewer 1 Report

The article is well written and well formulated and presents two theorems with their respective proofs and simulation results, which are very important for the presented formulations. The organization of the paper is good, the authors put some effort to present everything in a clear manner. However, there are some things that need to be addressed to meet the quality of publication. I would suggest major revision as follow:

1/ The novelty of the approach of the manuscript does not seem significant since it seems all the methods are existing and the experimental contribution does not consider.

2/ The motivation and background of wide practical use of the theoretic results presented should be clearly emphasized to facilitate the readers.

3/ In the introduction part, the literature survey is quite good. However, I think that the authors could enrich the reference section by discussing some new works related to sliding mode control and adaptive robust control methods, especially the dynamic sliding mode control methods, robust sliding mode method, multiple sliding mode methods and so on, should be included. To help the authors in this direction, I suggest the following reference: perturbation observer-based robust control using a multiple sliding surfaces for nonlinear systems with influences of matched and unmatched uncertainties, design of a non-singular adaptive integral-type finite time tracking control for nonlinear systems with external disturbances, finite-time convergence of perturbed nonlinear systems using adaptive barrier-function nonsingular sliding mode control with experimental validation, fast terminal sliding control of underactuated robotic systems based on disturbance observer with experimental validation, adaptive nonsingular terminal sliding mode control for performance improvement of perturbed nonlinear systems, robust tracking composite nonlinear feedback controller design for time-delay uncertain systems in the presence of input saturation, and the introduction should be added to do a better job of explaining the existing methods and why they are or are not valuable. What research gap did you find from previous researchers in your field (it is still partially described, but needs to be expanded and made clearer)?  Mention it in the Novelties section. It will improve the strength of the article.

4/ Please highlight which section(s) discusses each of the contributions. This way there will be cohesiveness in the manuscript contents.

5/ If the Lyapunov functions are chosen via the viewpoint of practical application, the authors should give some effective suggestions. What new modifications are introduced in the Lyapunov stability method?

6/ This paper considered the environmental disturbance; however, the study does not correlate these essential factors with the model. Thus, the model is not well developed yet. What is the principle of choosing these disturbances? Please clarify all.

7/ As mentioned in the title of the paper, however, there is no result robustness under the parametric uncertainties. Therefore, it would be interesting if the model is more stochastic with such disturbances. The author needs to give more detailed data references or results.

8/  Which is the sliding mode existence domain? In other words, what happens if different reference inputs and disturbance inputs are applied?

9/ The authors need to make a brief description of the controller parameters in Table 2, which are derived from references or experience? How are the parameters in the Table 2 determined? Please add more details of how the parameters of the controller are obtained. How the full-state constraints are chosen? It is better to explain how the values of the control parameters in the proposed method are adjusted? Whether these parameters are optimal for simulation results? More explanation and evidence should be given in detail.

10/ In simulation, the disturbances di1=5cos(0.1*pi*t-pi/3), di2=4cos(0.2*pi*t+pi/2), and di3=3cos(0.1*pi*t+pi/6) what is the principle of choosing these disturbances? Maybe these disturbances are too small compared to control inputs?

11/ The simulation and experimental results are missing the most important results such as the control input, the torque output of motor tau_T? These results that vary with the time in both scenarios need to be added in the paper. Moreover, more discussions should be given to clearly demonstrate the effectiveness of the obtained results. The explanations and analysis of simulation results should be enriched to show the validity of the data.  

12/ The readability and presentation of the work should be improved. Some Figures appear too light for a clear print and comprehension; they are quite grainy but this may be a function of the PDF process such as figures 4, 5, and 8. Please substantially improve their quality.

13/ The authors only simulate the controller, the analysis in this paper should be supported by experimental results. The authors should use practical systems to validate the proposed methods with experiment results. The validity of these relevant to applications is impossible to judge without experimental testing

14/ The contribution of conclusion is not mentioned clearly. Conclusion needs extended elaboration on the topic, results, lessons learned and future works. The authors need to re-write the conclusion again to show the objective and contribution of this paper clearly. Also, based on the ideas presented and the results obtained in this paper, the author should indicate the direction of further research in the conclusion section.

15/ The manuscript writing can be further polished with professional English, some typos and grammatical errors should be checked carefully, and some formatting problems that need to be modified.

Reviewer 2 Report

    Even there is no major technical issue, however the most critical issue consists in the lack of any significant contribution for a possible publication in this current form. This article contains several similar definitions and theorems (both in terms of theorem and proof), and proofs of theorems are simple and are based on the definitions of sentences and are free from any creativity or innovation. Also, examples whose results are not clear Throughout Theorems 1-2, it can be seen that the process of dealing with the issue investigated in the manuscript is complex. The authors are recommended to further elaborate where the main difficulties lie in terms of obtaining the presented results.

1)    In Introduction part, the motivation contributing to this study needs to be further emphasized. Also, there is no critical discussion.

2)    All assumptions and constraints should be discussed.

3)    3) Eq (27) and (46) are not clear.

4) The unique features of the proposed approaches and the main advantages of the results over others have to be clearly commented.

5) The authors should refer any further developments about the proposed approach and also refer any application of this strategy. Both topics should be discussion in the conclusion section.

6) The disadvantage or the limitation of the proposed method must be described in conclusion

Round 2

Reviewer 1 Report

Thank you for the revised manuscript. The authors addressed my comments very well and I highly recommend it for publication in the current form. 

Reviewer 2 Report

The article contains some novel part and the article has been revised according to reviewer's previous comment and suggestion. I have no further comment about this manuscript. I think that the paper deserves to be published.